# Denosumab Is Superior to Raloxifene in Lowering Risks of Mortality and Ischemic Stroke in Osteoporotic Women

**DOI:** 10.3390/ph16020222

**Published:** 2023-02-01

**Authors:** Ting-Chun Liu, Chien-Ning Hsu, Wen-Chin Lee, Shih-Wei Wang, Chiang-Chi Huang, Yueh-Ting Lee, Chung-Ming Fu, Jin-Bor Chen, Lung-Chih Li

**Affiliations:** 1Division of Nephrology, Department of Internal Medicine, Kaohsiung Chang Gung Memorial Hospital and Chang Gung University College of Medicine, Kaohsiung 833, Taiwan; 2Department of Pharmacy, Kaohsiung Chang Gung Memorial Hospital and Chang Gung University College of Medicine, Kaohsiung 833, Taiwan; 3School of Pharmacy, Kaohsiung Medical University, Kaohsiung 807, Taiwan; 4Institute for Translational Research in Biomedicine, Kaohsiung Chang Gung Memorial Hospital and Chang Gung University College of Medicine, Kaohsiung 833, Taiwan

**Keywords:** cardiovascular disease, ischemic stroke, mortality, denosumab, raloxifene, osteoporosis

## Abstract

Both osteoporosis and cardiovascular disease (CVD) share similar pathways in pathophysiology and are intercorrelated with increased morbidity and mortality in elderly women. Although denosumab and raloxifene are the current guideline-based pharmacological treatments, their impacts on cardiovascular protection are yet to be examined. This study aimed to compare mortality rate and cardiovascular events between denosumab and raloxifene in osteoporotic women. Risks of CVD development and all-cause mortality were estimated using Cox proportional hazard regression. A total of 7972 (3986 in each group) women were recruited between January 2003 and December 2018. No significant difference between denosumab and raloxifene was observed in composite CVDs, myocardial infarction, or congestive heart failure. However, comparison of the propensity score matched cohorts revealed that patients with proportion of days covered (PDC) ≥60% had lower incidence of ischemic stroke in the denosumab group than that in the raloxifene group (aHR 0.68; 95% CI 0.47–0.98; *p* = 0.0399). In addition, all-cause mortality was lower in the denosumab group than in the raloxifene group (aHR 0.59; 95% CI 0.48–0.72; *p* = 0.001), except in patients aged <65 y/o in this cohort study. We concluded that denosumab is superior to raloxifene in lowering risks of all-cause mortality and certain ischemic strokes in osteoporotic women.

## 1. Introduction

Osteoporosis and resultant fracture have become a public health problem, leading to a heavy global economic, social, and health burden. A previous study reported the estimation that 200 million women worldwide suffer from osteoporosis [1]. In particular, global deaths and disability-adjusted life-years attributable to osteoporosis and its related fracture increased from 207,367 and 8,588,936 in 1990 to 437,884 and 16,647,466 in 2019, respectively [2]. Aging has a major impact on the arterial system and heart, leading to an increase in cardiovascular disease (CVD) including atherosclerosis, hypertension, myocardial infarction, and stroke [3]. Both osteoporosis and CVD events are associated with physical disability, higher health care costs, impaired quality of life, and increased mortality [4,5]. The relationship between osteoporosis and CVD could be explained by their common risk factors such as age, smoking, alcohol consumption, physical activity, and menopause. Recent evidence has shown that osteoporosis and CVD share similar pathophysiological pathways and that osteoporotic patients are at a higher risk of developing major CVDs, such as stroke and ischemic heart disease [6].

The receptor activator of nuclear factor-κB ligand (RANKL)/the receptor activator of nuclear factor-κB (RANK)/osteoprotegerin (OPG) is one of the key signaling pathways shared by osteoporosis and CVD. RANKL–RANK interaction on the surface of pre-osteoclasts that activates downstream signals that initiate osteoclast activation, differentiation, and function [7]. OPG acts as a decoy receptor for RANKL, preventing RANK–RANKL interactions and blocking the resulting downstream osteoclastogenic cascade, and exerts an anti-calcific effect within the vasculature [8]. Denosumab is a fully human monoclonal antibody against human RANKL that mimics OPG, thus preventing it from activating RANK, and inhibits osteoclastogenesis, thereby decreasing bone resorption and increasing bone density. According to the Fracture Reduction Evaluation of Denosumab in Osteoporosis Every 6 Months (FREEDOM) trial, denosumab reduces the risk of vertebral, nonvertebral, and hip fractures. It can increase bone mineral density (BMD) at the lumbar spine and the total hip [9]. Treatment with a recombinant fusion protein, Fc-OPG, has been shown to inhibit vascular calcification in animal studies [10]. A study conducted with human RANKL knock-in (huRANKL-KI) mice showed that treatment with denosumab reduced aortic calcium deposits in prednisolone-treated huRANKL-KI mice by up to 50% based on calcium measurement [11]. However, no related study was conducted in humans. Considering the above information, we hypothesized that denosumab may have an influence in reducing cardiovascular events by slowing the progression of vascular calcification and decreasing the risk of CVD or heart failure.

Estrogen deprivation is associated with decreased bone mass and increased osteoclast formation [12,13,14]. Raloxifene is a benzothiophene non-steroidal derivative that is used as a second-generation selective estrogen receptor modulator (SERM). It binds to estrogen receptor and produces estrogen-like effects on bone, reducing resorption and increasing BMD in postmenopausal women. Raloxifene also acts as an estrogen agonist in pre-osteoclasts, inhibiting their proliferative capacity [15]. The Multiple Outcomes of Raloxifene Evaluation (MORE) trial, the pivotal treatment trial of raloxifene, demonstrated significant reductions in the risk of vertebral fractures after three years. [16]. Although raloxifene therapy for four years did not adversely affect the risk of cardiovascular events in the overall cohort, it reduced the risk of cardiovascular events in the subset of women with increased cardiovascular risk [17].

Denosumab has been shown to exert protective effects against CVD [18]. Chen et al. [19] reported that denosumab suppress the progression of coronary arterial calcification. Raloxifene may offer cardiovascular protection by normalizing the lipid profile, reducing oxidative stress, and improving endothelial function via increased nitric oxide production [20,21,22]. A recently published meta-analysis of randomized controlled trials demonstrated favorable effects of raloxifene on the lipid profile in women [23,24]. However, no direct comparison of the mortality and cardiovascular protective effects between denosumab and raloxifene has been made. As denosumab and raloxifene are currently recommended in selected patients by several osteoporosis management guidelines [25,26,27,28], it is urgently necessary to obtain evidence of these two medications regarding their cardiovascular benefits. The present study aimed to compare the effects of denosumab and raloxifene on mortality and CVD prevention.

## 2. Results

### 2.1. Patient Characteristics

A total of 33,576 adult osteoporotic patients receiving denosumab or raloxifene were identified, of whom 30,434 met the inclusion criteria (denosumab: 16,754 patients; raloxifene: 13,680 patients) (Figure 1).

Compared with patients initiated with raloxifene, those initiated with denosumab were older (73.18 ± 9.72 vs. 69.58 ± 10.76 years old) and had more comorbidities, including history of dementia, liver diseases, diabetes, renal diseases, hypertension, hyperlipidemia, and obstructive sleep apnea (Table 1). In the propensity score matched (PSM) cohort, 7972 denosumab- and raloxifene-matched pairs were analyzed over a six-year follow-up period. The baseline characteristics were well balanced in the matched groups and are presented in Table 1. Approximately 45% of the study cohort had a history of hypertension, 27% of hyperlipidemia, and 23% of diabetes. During the follow-up period, most patients were concomitantly administered medications for calcium (29%)/VitD (21%), followed by antihypertensive agents ACEi/ARB/Aliskiren (27%), statins (20%), anti-diabetics (18%), and thrombotic prevention. The frequency of these medications was similar between the denosumab and raloxifene groups (Table 1).

### 2.2. CVD Incidence

During the six-year follow-up period, the incidence of all CVDs was 7.80% (n = 311) and 10.71% (n = 427), and the incidence of all-cause mortality was 4.47% (n = 178) and 8.10% (n = 323) in the denosumab and raloxifene groups, respectively. Among the major CVDs, the highest event rate was ischemic stroke (4.59% in denosumab, 7.12% in raloxifene), followed by congestive heart failure (3.24% vs. 4.04% in denosumab vs. raloxifene groups) and myocardial infarction (1.00% vs. 1.56% in denosumab vs. raloxifene groups) (Table 2).

The cumulative incidence of mortality and CVDs between the denosumab and raloxifene groups is illustrated in Figure 2. No significant difference was observed in composite CVD incidence (Figure 2a: log-rank test, *p* = 0.3362) and heart failure (Figure 2c: log-rank test, *p* = 0.4819) between the two groups. Notably, the incidence of ischemic stroke was significantly lower in the denosumab group than in the raloxifene group (Figure 2b; log-rank test, *p* = 0.0304).

Following adjustment for the baseline characteristics, comorbidities, and the use of concomitant medications, no significant difference was observed between denosumab and raloxifene in MI, ischemic stroke, heart failure, and all-cause CVDs. However, in the stratified analyses, we found that among patients with proportion of days covered (PDC) ≥60%, the incidence of ischemic stroke was significantly lower in the denosumab group than in the raloxifene group (adjusted hazard ratio (aHR) 0.68; 95% CI 0.47–0.98; *p* = 0.0399) (Figure 3).

### 2.3. All-Cause Mortality

Of note, all-cause mortality was lower in the denosumab group than in the raloxifene group. Kaplan–Meier analysis showed that the all-cause mortality was significantly lower in the denosumab group than in the raloxifene group (Figure 2d; log-rank test, *p* = 0.0022). The stratified analyses further confirmed the superiority of denosumab to raloxifene in lowering the risk of all-cause mortality (aHR 0.59; 95% CI 0.48–0.72; *p* = 0.001). In patients less than 65 years of age, denosumab users showed lower all-cause mortality, though the difference was not significant. Among osteoporotic women with varying degrees of renal impairment, the superiority of denosumab to raloxifene in lowering the risk of all-cause mortality was apparent (Figure 3).

## 3. Discussion

Multimorbidity is a growing concern, especially in elderly patients and patients with chronic diseases [29,30,31,32,33]. Organ protection therefore emerges as a key concept in the management of chronic diseases including hypertension, diabetes, and chronic kidney disease. Osteoporosis and resultant fracture have been well-recognized as a silent threat to elderly patients and postmenopausal women. Both osteoporosis and CVD are common in these patient populations. As CVD and osteoporosis share some pathophysiological similarities, we quantified the cardiovascular benefits of denosumab and raloxifene based on the concept of organ protection.

Denosumab and raloxifene are recommended for osteoporosis treatment in postmenopausal women. Dose adjustment of both medications is not required in renal function impairment. In the present study, we showed that denosumab is superior to raloxifene in reducing the risk of ischemic stroke, in particular, in patients with PDC ≥60% and with eGFR 30~59.9 mL/min/1.73 m^2^. In line with our findings, the results of the Raloxifene Use for The Heart (RUTH) trial revealed that raloxifene may carry increased risks of venous thromboembolism and fatal stroke [34].

Comparative event rates of MI or heart failure in the denosumab and raloxifene groups were found in our study. Raloxifene has been reported to improve the levels of serum low-density lipoprotein cholesterol, total cholesterol, and non-high-density lipoprotein cholesterol in the post-hoc analysis of the MORE trial [35]. However, in the RUTH trial, a large randomized controlled trial, raloxifene did not significantly affect the risk of coronary heart disease [34]. On the other hand, in recent years, the RANKL/RANK/OPG pathway has been discovered not only as a regulator of bone remodeling but also as a regulator of vascular calcification [36]. Despite the fact that Samelson et al. [37] reported that denosumab had no effect on the progression of aortic calcification or the incidence of CV adverse events in postmenopausal women, 24% patients did not finish complete doses of denosumab, and the follow-up period was only three years. Recently, Suzuki et al. [38] found that denosumab ameliorated aortic arch calcification in dialysis patients. The calcified area apparently diminished through 30 months of treatment and it was deemed necessary for ≥2 years for decalcification due to denosumab. Hsu et al. [18] demonstrated that denosumab use is associated with a significant reduction in cardiovascular events compared to alendronate use with better adherence of >60% medication possession ratio. Despite no significant differences in MI and heart failure between the denosumab and raloxifene groups, our study did not obtain evidence of vascular calcification in X-ray or CT, owing to data limitations. Further study is necessary to elucidate whether denosumab can improve vascular calcification.

We demonstrated the lower risk of all-cause mortality in the denosumab group compared with the raloxifene group. Effective osteoporosis treatment has been reported to reduce mortality, per se, because in addition to decreasing bone fracture, treating osteoporosis might improve the ability of an individual to cope with or recover from an acute illness, probably by maintaining physiological reserve and preventing frailty [39]. Head-to-head comparison of denosumab and raloxifene in various bone fractures is lacking. However, denosumab was proven to effectively reduce risks of vertebral, non-vertebral, and hip fracture in the FREEDOM trail [9], whereas raloxifene only reduced the risk of vertebral fracture in the MORE trial [16]. Wu et al. also demonstrated that high adherence users of denosumab, defined by receiving three or four doses, had lower risks of all-cause mortality than low adherence users of denosumab, defined by one or two doses (aHR 0.64, 95% CI 0.48–0.86) [40]. Moreover, the RANKL/RANK/OPG pathway plays numerous roles in various organs [41]. This pathway has been shown to participate in intestinal immunity [42], central nervous system inflammation [43], skin inflammation [44], and fever that occurs during infection [45]. This preclinical evidence supports the potential benefit of denosumab treatment in patient populations suffering from various infectious and inflammatory diseases. Although these potential benefits might explain the lower risks of all-cause mortality shown in our study, additional clinical investigations are required to examine the therapeutic benefit of denosumab in these settings.

Although we performed analysis using one of the largest and most representative medical research databases in Taiwan [46], there are certain limitations to the current study. First, from the database studied, we were unable to identify specific causes of death in individual patients. Second, there were certain unmeasured confounding variables, such as smoking, alcohol, and home nutrient supplement. Finally, laboratory results, such as 1,25(OH)2 vitamin D and parathyroid hormone, are not part of routine care nor health insurance imbursed tests. Despite these limitations, the present study was valuable because we demonstrated the differences in lowering risks of all-cause mortality and ischemic stroke in denosumab compared to raloxifene in osteoporotic women.

## 4. Materials and Methods

### 4.1. Data Source

This retrospective cohort study was conducted using electronic health record (EHR) data from the Chang Gung Research Database (CGRD), an electronic healthcare delivery system derived from a group of Chang Gung Memorial Hospitals (CGMH) in geographically distinct parts of Taiwan. Briefly, CGMH is the largest healthcare delivery system in Taiwan, providing approximately 10–12% of the healthcare services of the Taiwan National Health Insurance (NHI) program [46]. The Taiwan NHI program is a compulsory, single-payer health insurance program that covers over 99% of Taiwan’s population [47]. The CGRD, containing detailed individual patient-level EHR including diagnosis, prescription, and laboratory test results, has been validated in some disease populations [46,48]. We conducted this analysis using the CGRD data from January 2003 to December 2018.

### 4.2. Study Design and Study Cohort

By using the EHR database, we first identified patients aged 30–89 years at the time of initiation of denosumab or raloxifene between 1 January 2003 and 31 December 2017 to be included in the study. To develop the new user cohort, only patients with ≥1-year records before treatment initiation were included in the study. To assess the risk of CVD, patients were excluded with history of stroke, myocardial infarction, heart failure, and cancer. Patients receiving procedures for cardiovascular diseases, including percutaneous coronary intervention (PCI) and coronary artery bypass surgery (CABG), were also identified before treatment initiation and excluded from analyses. Because raloxifene was used in women, male patients were excluded from the analysis. Operational definitions and codes for disease conditions and procedures are listed in Appendix A.

### 4.3. Comparison Groups

New users of denosumab were defined as patients who never taken raloxifene within one year before treatment initiation, and the earliest date of denosumab prescribed was defined as the index date. The same criteria were applied for new users of raloxifene without prior denosumab treatment. PSM was applied to balance differences in baseline demographic and clinical characteristics between the denosumab and raloxifene groups. The propensity score of initiating denosumab or raloxifene was estimated by logistic regression with above mentioned baseline characteristics, with a 1:1 ratio.

### 4.4. Outcomes and Follow-Up

The outcomes of interest were composite of major CVD, including myocardial infarction, congestive heart failure, and ischemic stroke. These outcomes were assessed based on hospital discharge diagnoses. Individual outcomes of composite CVD were also analyzed.

### 4.5. Study Covariates

Baseline variables considered in the analyses included patient demographics; comorbid conditions, such as the Charlson Comorbid Index [49]. The following study covariates were measured at baseline as potential confounders to be adjusted in the analyses: procedures of PCI and CABG; preexisting hyperlipidemia; and prior medication use: anti-thrombotic medications (anti-coagulants, anti-platelets), aspirin, glucose-lowering agents, lipid-lowering agents, antihypertensive agents, vitamin D, or calcium supplementations, as well as other osteoporosis therapy besides study drugs.

### 4.6. Statistical Analysis

Data were summarized as mean ± standard deviation (SD) for continuous variables and n (%) for categorical variables in the study cohort. PSM was employed to balance the differences in baseline demographic and clinical characteristics between the denosumab and raloxifene groups. New users of denosumab and raloxifene were matched at a 1:1 ratio using the greedy algorithm of the PSM method. The covariate balance between the denosumab and raloxifene groups was measured using the standardized mean difference (SDM), and an SDM of <0.1 was considered as no meaningful difference [50].

We used the Cox proportional model to estimate the aHR of the study outcomes between the new users of denosumab and raloxifene. Time to CVD endpoint was used using Kaplan–Meier analysis with log-rank tests. Cox proportional hazard regression was performed for composite incident CVD events and individual cardiovascular events.

Importantly, to assess the heterogeneous effects of denosumab (versus raloxifene) with different baseline characteristics, stratified analyses were performed in the matched cohorts by age < 65 years (vs. age ≥ 65 years) and baseline eGFR groups (≥60, 30–59.9, and <30 mL/min/1.73 m^2^). A two-tailed test (*p* value < 0.05) was considered statistically significant. All statistical analyses were performed using SAS 9.4 (SAS Institute, Cary, NC, USA).

## 5. Conclusions

In conclusion, denosumab is superior to raloxifene in lowering risks of all-cause mortality and certain ischemic strokes in osteoporotic women. Taken into the concept of organ protection in the pharmacological treatment of osteoporosis, denosumab could be a better choice than raloxifene.

## Figures and Tables

**Figure 1 pharmaceuticals-16-00222-f001:**
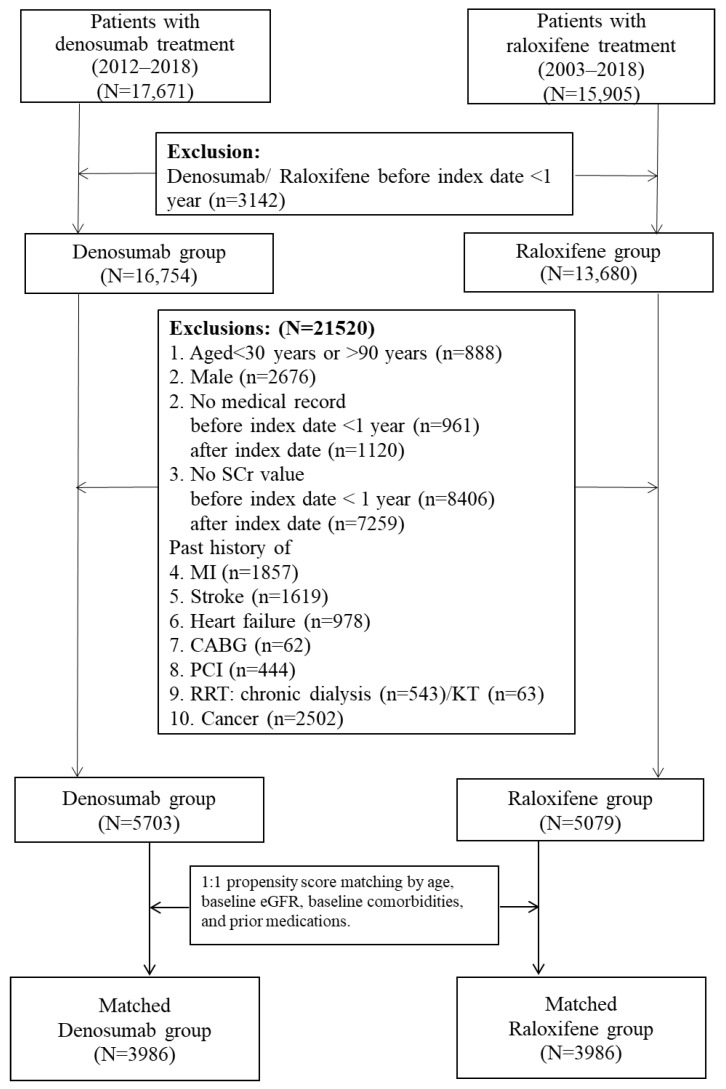
Study flowchart and patient selection process. Propensity score was calculated based on baseline comorbidities: peripheral vascular diseases, dementia, pulmonary disease, connective ti-sue disorder, peptic ulcer, liver diseases, diabetes, diabetes complications, paraplegia, renal disease, severe liver diseases, metastatic cancer, hypertension, hyperlipidemia, abnormal thyroid function, obstructive sleep apnea, and prior medication (oral anticoagulants, antiplatelets, aspirin, statins, fibrates, other lipid-lowering agents, anti-diabetics, angiotensin-converting enzyme inhibitors (ACEI)/angiotensinreceptor blockers (ARBs)/Aliskiren, diuretics, bisphosphonates, Forteo, and calcitonin preparations). CABG, coronary artery bypass surgery; MI, myocardial infarction; PCI, percutaneous coronary intervention; RRT, renal replacement therapy; and SCr, serum creatinine.

**Figure 2 pharmaceuticals-16-00222-f002:**
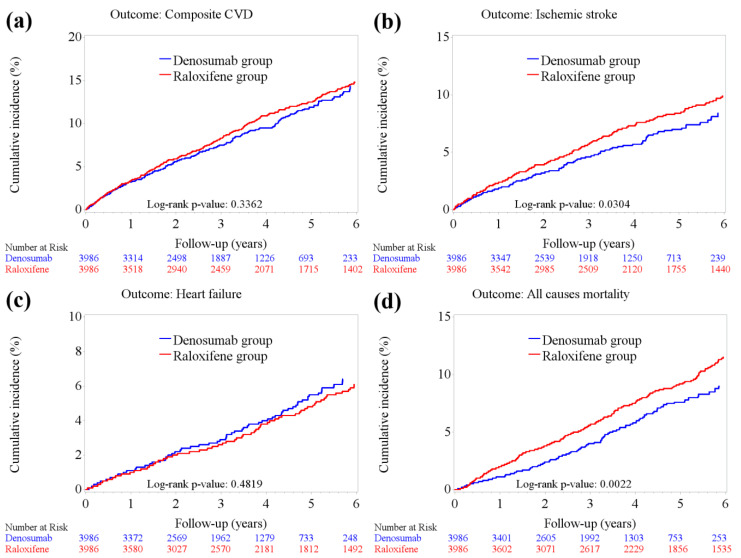
Cumulative incidence of mortality and cardiovascular outcomes in the denosumab and raloxifene groups. (**a**) Composite cardiovascular (CVD) events (log-rank test, *p* = 0.3362); (**b**) ischemic stroke (log-rank test, *p* = 0.0304); (**c**) heart failure (log-rank test, *p* = 0.4819); and (**d**) all-cause mortality (log-rank test, *p* = 0.0022).

**Figure 3 pharmaceuticals-16-00222-f003:**
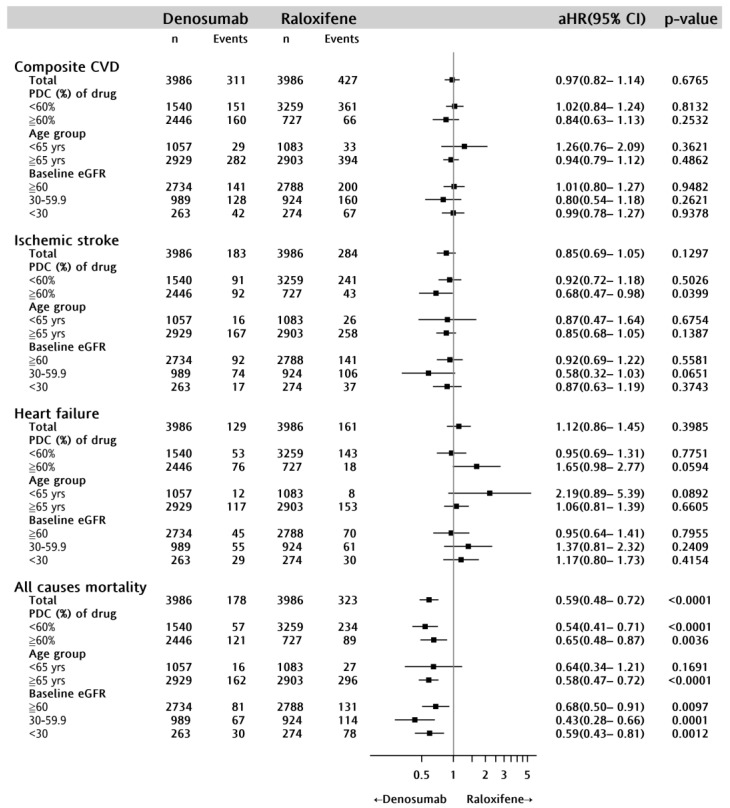
Stratified analyses of the risk of composite CVD, ischemic stroke, heart failure, and all–cause mortality between the denosumab and raloxifene groups (denosumab versus raloxifene). CVD, cardiovascular disease; aHR, adjusted hazard ratio; PDC, proportion of days covered; and eGFR, estimated glomerular filtration rate. # Adjusted for age, sex, baseline eGFR, mean dose of drug, CVD-related comorbidities, baseline comorbidities (abnormal thyroid function, obstructive sleep apnea, and fracture), and concomitant medications (anti-thrombotic agents, lipid-lowering agents, antidiabetics, antihypertensive agents, and other osteoporosis therapy).

**Table 1 pharmaceuticals-16-00222-t001:** Characteristics of the study cohorts before and after PSM.

	Before PSM							After PSM						
		Denosumab (n = 5703)		Raloxifene (n = 5079)					Denosumab (n = 3986)		Raloxifene (n = 3986)			
	N	n	(%)	n	(%)	SMD	*p-*Value	N	n	(%)	n	(%)	SMD	*p-*Value
Age at the index date (years), mean (SD)	10,782	73.18	±9.72	69.58	±10.76	0.35	<0.0001	7972	71.31	± 9.97	71.32	± 10.00	0.00	0.9587
Baseline eGFR, mean (SD)	10,782	76.07	±30.95	73.01	±29.44	0.10	<0.0001	7972	74.11	± 29.56	74.29	± 30.16	0.01	0.7977
Baseline comorbidities														
Peripheral vascular diseases	154	90	(1.58)	64	(1.26)	0.03	0.1648	109	54	(1.35)	55	(1.38)	0.00	0.9232
Dementia	464	300	(5.26)	164	(3.23)	0.10	<0.0001	320	165	(4.14)	155	(3.89)	0.01	0.5683
Pulmonary disease	1084	618	(10.84)	466	(9.18)	0.06	0.0042	783	387	(9.71)	396	(9.93)	0.01	0.7348
Connective tissue disorder	527	301	(5.28)	226	(4.45)	0.04	0.0465	393	196	(4.92)	197	(4.94)	0.00	0.9587
Peptic ulcer	2227	1160	(20.34)	1067	(21.01)	0.02	0.3925	1659	841	(21.10)	818	(20.52)	0.01	0.5257
Liver diseases	1291	739	(12.96)	552	(10.87)	0.06	0.0008	942	464	(11.64)	478	(11.99)	0.01	0.6271
Diabetes	2520	1422	(24.93)	1098	(21.62)	0.08	<0.0001	1851	930	(23.33)	921	(23.11)	0.01	0.8113
Diabetes complications	756	445	(7.80)	311	(6.12)	0.07	0.0007	542	273	(6.85)	269	(6.75)	0.00	0.8587
Paraplegia	64	28	(0.49)	36	(0.71)	0.03	0.1416	43	24	(0.60)	19	(0.48)	0.02	0.4445
Renal disease	1007	623	(10.92)	384	(7.56)	0.12	<0.0001	687	345	(8.66)	342	(8.58)	0.00	0.9047
Severe liver diseases	41	22	(0.39)	19	(0.37)	0.00	0.9217	31	12	(0.30)	19	(0.48)	0.03	0.2078
Metastatic cancer	4	3	(0.05)	1	(0.02)	0.02	0.3757	2	1	(0.03)	1	(0.03)	0.00	1.0000
Hypertension	5058	2911	(51.04)	2147	(42.27)	0.18	<0.0001	3626	1822	(45.71)	1804	(45.26)	0.01	0.6856
Hyperlipidemia	2991	1760	(30.86)	1231	(24.24)	0.15	<0.0001	2154	1085	(27.22)	1069	(26.82)	0.01	0.6865
Thyroid function abnormal	288	176	(3.09)	112	(2.21)	0.05	0.0046	183	84	(2.11)	99	(2.48)	0.03	0.2620
Obstructive sleep apnea	269	204	(3.58)	65	(1.28)	0.15	<0.0001	136	72	(1.81)	64	(1.61)	0.02	0.4890
Prior medications														
Oral anticoagulants	175	121	(2.12)	54	(1.06)	0.08	<0.0001	98	48	(1.20)	50	(1.25)	0.00	0.8389
Anti-platelet	1536	858	(15.04)	678	(13.35)	0.05	0.0119	1154	597	(14.98)	557	(13.97)	0.03	0.2029
Aspirin	1183	667	(11.70)	516	(10.16)	0.05	0.0109	886	460	(11.54)	426	(10.69)	0.03	0.2257
Statins	2244	1368	(23.99)	876	(17.25)	0.17	<0.0001	1605	818	(20.52)	787	(19.74)	0.02	0.3866
Fibrates	225	147	(2.58)	78	(1.54)	0.07	0.0002	143	72	(1.81)	71	(1.78)	0.00	0.9328
Anti-diabetics	2033	1143	(20.04)	890	(17.52)	0.06	0.0008	1490	750	(18.82)	740	(18.56)	0.01	0.7739
ACEI/ARB/Aliskiren	3046	1794	(31.46)	1252	(24.65)	0.15	<0.0001	2162	1096	(27.50)	1066	(26.74)	0.02	0.4498
Diuretics	518	217	(3.81)	301	(5.93)	0.10	<0.0001	368	186	(4.67)	182	(4.57)	0.00	0.8309
Alendronate	1431	770	(13.50)	661	(13.01)	0.01	0.4566	1045	511	(12.82)	534	(13.40)	0.02	0.4453
Forteo (Teriparatide)	279	160	(2.81)	119	(2.34)	0.03	0.1310	226	116	(2.91)	110	(2.76)	0.01	0.6856
Calcium	3192	1363	(23.90)	1829	(36.01)	0.27	<0.0001	2343	1157	(29.03)	1186	(29.75)	0.02	0.4759
Vit. D	2264	945	(16.57)	1319	(25.97)	0.23	<0.0001	1698	837	(21.00)	861	(21.60)	0.01	0.5115

Abbreviations: ACEI, angiotensin converting enzyme inhibitor; ARB: angiotensin receptor blockers; PSM, propensity score matching; SMD, standardized mean difference. SMD < 0.1 was considered as no sign of imbalance.

**Table 2 pharmaceuticals-16-00222-t002:** The major adverse cardiovascular outcomes in the denosumab and raloxifene groups.

	N	Denosumab (n = 3986)		Raloxifene (n = 3986)		*p*-Value
		n	(%)	n	(%)	
Primary outcomes						
Any CVD	738	311	(7.80)	427	(10.71)	<0.0001
Myocardial infarction	102	40	(1.00)	62	(1.56)	0.0284
Ischemic stroke	467	183	(4.59)	284	(7.12)	<0.0001
Congestive heart failure	290	129	(3.24)	161	(4.04)	0.0556
In-hospital death from any cause	501	178	(4.47)	323	(8.10)	<0.0001

## Data Availability

Data is contained within the article and Appendix A.

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
