# Peer review of "Denosumab Is Superior to Raloxifene in Lowering Risks of Mortality and Ischemic Stroke in Osteoporotic Women"

_pharmaceuticals, 2023, doi:10.3390/ph16020222_

Round 1

Reviewer 1 Report

By their manuscript “Denosumab is Superior to Raloxifene in Lowering Risks of Mortality and Ischemic Stroke in Osteoporotic Women” Ting-Chun Liu and co-authors continue a series of papers evaluating the various kidney and cardiac outcomes of Denosumab and Raloxifene users in women with osteoporosis. In particular, in the present paper the authors evaluated the effects of Denosumab on mortality and cardiovascular disease prevention compared to those of Raloxifene. The paper is written properly and addresses correctly the topic. The results support the conclusions, even considering the methodological limitations of the study. Since the results are of interest and the concept potentially of scientific value, I suggest acceptance of the manuscript in its present form.

Author Response

We thank the reviewer for your comments. We have uploaded the revised manuscript, in which we have added more description in the mothods section (labled in red). We also improved the readability of our manuscript by rephrasing the words. (labled in blue)

Reviewer 2 Report

The work titled: Denosumab is Superior to Raloxifene in Lowering Risks of Mortality and Ischemic Stroke in Osteoporotic Women, well-thought out and with the right methodology. I recommend the publication as in present form.

Author Response

We thank the reviewer for your comments. We have uploaded the revised version of manuscript, in which we have added more description in the methods section (labled in red). We also improved the readability of our manuscript by rephrasing the words. (labled in blue)

Reviewer 3 Report

very Respected Authors,

After carefully reading your manuscript I have few suggestions.

What type of the study you have used? Please describe methodology more. The approval of the Ethical Committee stands in the section Method, at the end, after the subsection Study Design. The number and the date of approval is necessary.

Author Response

We thank the reviewer for your comments. This study is a retrospective cohort study using electronic health record (EHR) data from the Chang Gung Research Database. We have uploaded the revised manuscript, in which we have added more description in the methods section (labled in red). We have already added the approval date of IRB in the Institutional Review Board Statement and deleted the sentence in methods section. We also improved the readibility of our manuscript by rephrasing the words. (labled in blue)
